# Antisense Oligonucleotide-Based Therapeutic against Menin for Triple-Negative Breast Cancer Treatment

**DOI:** 10.3390/biomedicines9070795

**Published:** 2021-07-08

**Authors:** Dang Tan Nguyen, Thi Khanh Le, Clément Paris, Chaïma Cherif, Stéphane Audebert, Sandra Oluchi Udu-Ituma, Sébastien Benizri, Philippe Barthélémy, François Bertucci, David Taïeb, Palma Rocchi

**Affiliations:** 1Predictive Oncology Laboratory, Centre de Recherche en Cancérologie de Marseille (CRCM), Inserm UMR 1068, CNRS UMR 7258, Institut Paoli-Calmettes, Aix-Marseille University, 27 Bd. Leï Roure, 13273 Marseille, France; nguyendangtancdy@gmail.com (D.T.N.); khanh.le-thi@inserm.fr (T.K.L.); clement.paris@inserm.fr (C.P.); chayma.che@gmail.com (C.C.); uduitumasandra@gmail.com (S.O.U.-I.); bertuccif@ipc.unicancer.fr (F.B.); david.taieb@ap-hm.fr (D.T.); 2Department of Life Science, University of Science and Technology of Hanoi (USTH), Hanoi 000084, Vietnam; 3Marseille Protéomique, Centre de Recherche en Cancérologie de Marseille, INSERM, CNRS, Institut Paoli-Calmettes, Aix-Marseille University, 13009 Marseille, France; stephane.Audebert@inserm.fr; 4ARNA Laboratory, INSERM U1212, CNRS UMR 5320, University of Bordeaux, 33076 Bordeaux, France; sebastien@benizri.fr (S.B.); philippe.barthelemy@inserm.fr (P.B.); 5Biophysics and Nuclear Medicine Department, La Timone University Hospital, European Center for Research in Medical Imaging, Aix-Marseille University, 13005 Marseille, France

**Keywords:** menin, triple-negative breast cancer (TNBC), antisense oligonucleotides, apoptosis, interactome

## Abstract

The tumor suppressor menin has dual functions, acting either as a tumor suppressor or as an oncogene/oncoprotein, depending on the oncological context. Triple-negative breast cancer (TNBC) is characterized by the lack of expression of the estrogen receptor (ER), progesterone receptor (PR), and human epidermal growth factor receptor 2 (ERBB2/HER2) and is often a basal-like breast cancer. TNBC is associated with a dismal prognosis and an insufficient response to chemotherapies. Previously, menin was shown to play a proliferative role in ER-positive breast cancer; however, the functions of menin in TNBC remain unknown. Here, we have demonstrated that menin is expressed in various TNBC subtypes with the strongest expression in the TNBC Hs 578T cells. The depletion of menin by an antisense oligonucleotide (ASO) inhibits cell proliferation, enhances apoptosis in Hs 578T cells, highlighting the oncogenic functions of menin in this TNBC model. ASO-based menin silencing also delays the tumor progression of TNBC xenografts. Analysis of the menin interactome suggests that menin could drive TNBC tumorigenesis through the regulation of MLL/KMT2A-driven transcriptional activity, mRNA 3′-end processing and apoptosis. The study provides a rationale behind the use of ASO-based therapy, targeting menin in monotherapy or in combination with chemo or PARP inhibitors for menin-positive TNBC treatments.

## 1. Introduction

Worldwide, breast cancer is the most common cancer in women [1] (GLOBOCAN 2020, https://gco.iarc.fr/, accessed on 30 December 2020). At diagnosis, one-third of cases have regional lymph node extension and less than 10% have distant metastases. The choice of the treatment heavily depends on tumor stage, ranging from curative to palliative approaches. In the era of precision medicine, the treatment is guided by molecular genetics but validated targets remain limited in number [2].

The main molecular targets in breast cancer pathogenesis are estrogen receptor alpha (ERα), which is expressed in approximately 75% of invasive breast cancers and human epidermal growth factor receptor 2 (ERBB2/HER2), amplified or overexpressed in approximately 15% of breast cancers [3]. There are three major breast cancer subtypes: luminal ER positive (including luminal A and luminal B), ERBB2 enriched and basal-like or triple-negative [4]. Triple-negative breast cancer (TNBC) is immunohistochemically characterized by negative expression of estrogen receptors (ER), progesterone receptors (PR), and HER2 and often a basal-like breast cancer (BLBC) [5,6,7]. However, TNBC remains a heterogeneous group of tumors with at least six gene expression profile-based subtypes [6,8,9]. TNBC is characterized by a worse overall survival, greater relapse rates, and higher metastatic potential than other breast cancer subtypes [6,7,8,10]. Due to the absence of ER, PR, HER2 expressions, TNBC tumors are unresponsive to hormone therapy or HER2-directed treatment, and TNBC patients also have limited susceptibility to chemotherapy; therefore, TNBC remains the most challenging breast cancer subtype to overcome [8].

The tumor suppressor menin, encoded by the *MEN1* gene can act as either a tumor suppressor or an oncogene/oncoprotein, depending on the oncological context [11]. Mutations in the *MEN1* gene lead to an inherited cancer syndrome named multiple endocrine neoplasia type 1 (MEN1, disease identifier OMIM131100) typically characterized by parathyroid, pancreatic and pituitary tumors [12,13]. In MEN1 disorder, for example, menin acts as a tumor suppressor as loss-of-function mutations in the *MEN1* gene are responsible for tumor developments of more than 95% of MEN1 patients [11]. Additionally, endocrine tumor development was observed in mice with heterozygous *MEN1* deletion, supporting the tumor suppressor role of menin [14]. By contrast, menin acts as an oncogenic cofactor of MLL (mixed lineage leukemia) fusion protein in aggressive lymphoid and myeloid leukemias, and the interruption of menin−MLL interaction can block the oncogenic function of the MLL fusion protein [15]. Of note, menin has a dual role in breast cancer as reported in previous studies [16]. Women with the MEN1 syndrome showed an increased risk of breast cancer occurrence, and heterozygous *MEN1* deletion or loss of menin expression were observed in breast cancer cells of MEN1 patients [17], indicating the tumor suppression function of menin in breast tumorigenesis. On the other hand, the proliferative activity of menin was found in ER-positive breast cancer cells [18,19]. Menin co-activates ERα through its direct interaction with ERα, and menin overexpression can compete with tamoxifen on the AF-2 binding site of ERα, leading to a resistance to therapy [19]. From a clinical standpoint, the analysis of expression by immunohistochemistry (IHC) study in a series of 65 ER-positive breast cancer cases treated with tamoxifen for 2–5 years, revealed that menin-positive patients have a worse prognosis than their menin-negative counterparts [19]. Menin participates in many cellular processes and has a critical role in the recruitment of nuclear receptor-mediated transcription factors [11]. In particular, menin acts as an integral part of the histone methyltransferase complex MLL/KMT2A/MLL2/KMT2D, which is involved in H3K4 trimethylation, an epigenetic marker of the transcription activity of menin [20,21,22]. Menin was found to interact with DNA via FOXA1/GATA3 bound enhancers, which are looped to promoters and are enriched for ERα DNA binding [16,23].

During the past decades, nucleic acid-based technologies (e.g., siRNA, ASO, mRNAs, etc.) have shown great potentials for modulating the expression of oncogenes and disease-causing genes or correcting aberrant alternative splicing, and are fully integrated into precision medicine in oncology and genetic disorders [24,25]. Inhibitors of the menin/MLL1 complex have been developed for various malignancies but have shown limited efficiency in solid cancers [26,27]. The direct and specific inhibition of menin would represent an alternative approach and is expected to interrupt the diversity of oncogenic functions of menin. To this purpose, we have developed antisense oligonucleotides (ASOs) against menin mRNAs. ASOs have several advantages over siRNAs, such as their sufficient cellular uptake without the need of transfection reagents, their easier synthesis and lower cost of production [25]. Moreover, ASOs have a longer history of clinical development with eight approved ASOs since 1998 while there have been two approved siRNAs so far [25].

Menin has a dual role in breast cancer; however, whether menin functions as a tumor suppressor or menin has a proliferative activity in TNBC is completely unknown. In this study, we have aimed to investigate the expression of menin in various TNBC cells and assess the effects of specific ASO-driven menin inhibition on apoptosis and tumor progression in vitro and in vivo using the TNBC Hs 578T cell model. We also present the map of menin-interacting proteins identified by IP coupled MS, suggesting putative menin functions in TNBC.

## 2. Materials and Methods

### 2.1. Cell Lines

A set of triple negative breast cancer (TNBC) cell lines have been used for evaluating menin expressions. The human TNBC cell line Hs 578T which was studied throughout the study was maintained in Dulbecco’s Modified Eagle’s Medium (Life Technologies SAS, Courtaboeuf, France), supplemented with 10% fetal bovine serum (FBS), 0.5% human insulin (2 mg/mL), and 1% D-glucose 45%, 1% Hepes 1 M, 1% Antibiotic-Antimycotic (100X), 1 mM Sodium pyruvate 100 mM, 1% amino acid (Life Technologies SAS, Courtaboeuf, France). The non-tumorigenic epithelial breast cell line MCF-10A ER, obtained from Thermo Fisher Scientific (Life Technologies SAS, Courtaboeuf, France), was maintained in medium containing DMEM/F-12 (Life Technologies SAS, Courtaboeuf, France ), supplemented with 5% fetal bovine serum (FBS) (Life Technologies SAS, Courtaboeuf, France ), 0.5% human insulin (2 mg/mL), 1% Hepes 1 M, 1% Antibiotic-Antimycotic 100X, 1% Sodium pyruvate (100X), 1% amino acid, 0.1% epidermal growth factor (EGF) (10 mg/mL), cholera toxin (100 mg/mL), 0.05% hydrocortisone (2 mg/mL). These cell lines were maintained at 37 °C in a 5% CO_2_ humidified atmosphere.

### 2.2. Design and Synthesis of Antisense Oligonucleotides

The menin-ASO sequence corresponding to the human *MEN1* mRNA at position 531–550 was 5′-ATGAAGCTGAAGAGGGACTG-3, whereas the scramble control sequence was 5′ CGTGTAGGTACGGCAGATC-3′ and designated as an ASO control.

To design the antisense oligonucleotides (ASOs) targeting the Menin mRNA, a program based on R that was created and patented by Finetti, P., Birnbaum, D., Bertucci, F and Rocchi, P. in our laboratory (PDA16130, 2017) was used. Firstly, the coding potion of the target transcript is selected and segmented into consecutive sequences of 20 bases. Subsequently, to define potential ASOs, the complementary sequences of the resulting sequences are identified and reversed to 5′–3′ direction. The program gives the list of ASO sequences with information regarding the GC content and genes showing significant similarity. Finally, the ASOs are selected basing on their percentage of GC and their specificity to the sequence of target transcripts. 

Oligonucleotides were synthesized on OligoPilot 10 automated DNA synthesizer (50 μmol scale). The oligonucleotide synthesis was carried out using standard β-cyanoethyl phosphoramidite chemistry. Oligonucleotide sequences were fully modified with phosphorothioate (PS) backbone. After synthesis, oligonucleotide cleavage and deprotection was carried out in concentrated ammonium hydroxide at 55 °C for 16 h. Purification was performed on ion pairing reversed-phase high-pressure liquid chromatography (IP-RP-HPLC). The purity was assessed by analytical IP-RP-HPLC and characterized by Maldi-Tof mass spectrometry.

### 2.3. Transfection with ASOs

For all of the ASO treatments, Hs 578T cells at 50–60% density were transfected twice with ASOs at 24 h and 48 h after cell seeding. ASOs at the indicated concentrations were pre-incubated for 20 min with 4 uL Oligofectamin (Life Technologies SAS, Courtaboeuf, France) in 1 mL Gibco Opti-MEM, a serum-reduced medium (Life Technologies SAS, Courtaboeuf, France) before adding to the cells. Following 4 h incubation, the medium containing ASOs and Oligofectamin was replaced with the completed medium. On the following day, the same procedure of ASO treatment was carried out. For Western blot analysis, the transfected cells were harvested 72 h after the second ASO transfection.

In order to enhance significantly cell uptake and cellular trafficking of ASOs during ASOs treatment, Oligofectamin (a cationic lipid- transfection reagent) was used. ASO transfections were performed twice to increase the ASOs internalized into the cells and increase its efficiency.

### 2.4. Western Blot

Protein extracts were prepared using the lysis buffer (1% *v*/*v* Triton X-100, 50 mM HEPES, 150 mM NaCl, 25 mM NaF, 1 mM EDTA, 1 mM EGTA, 10 μM ZnCl_2_, 1 mM sodium orthovanadate) containing 4% *v*/*v* protease inhibitor cocktail (Sigma Aldrich Chimie S.a.r.l, St. Quentin Fallavier, France) for 30 min on ice. The lysate was centrifuged (60 min, 13,000 rpm, 4 °C) and protein content was quantified using the BCA protein assay kit (Life Technologies SAS, Courtaboeuf, France). Samples of 40 μg protein were mixed with Laemmli sample buffer (LB4X), boiled for 5 min at 95 °C before being separated by SDS-polyacrylamide gels for electrophoresis. Proteins were transferred to PVDF membranes, which were blocked with 5% *w*/*v* nonfat milk in tris-buffered saline (TBS 1X) for 1 h. For immunodetection, the membranes were incubated with 1:500 mouse anti-MENIN monoclonal antibody (SC-390345, Santa Cruz Biotechnology, Heidelberg, Germany), 1:2000 rabbit anti-menin polyclonal antibody (Bethyl Laboratories Inc., Montgomery, TX, USA), 1/1000 rabbit anti-caspase-3 monoclonal antibody (9662s, Abcam, Cambridge, UK), 1/1000 rabbit anti- PARP (9542, Cell Signaling Technology, Beverly, MA, USA), 1/2500 rabbit anti-GAPH polyclonal antibody (Abcam, Cambridge, UK) as endogenous loading controls. Visualization of the protein bands was accomplished using Western Chemiluminescent HRP Substrate (GE Healthcare, Buckinghamshire, UK).

### 2.5. Cell Viability with Alamar Blue

Hs 578T cells were plated in 12-well plates (3 × 10^4^ cells/well) and transfected the day after with menin-ASO or control-ASO at 100 nM, 200 nM, 300 nM and 400 nM. After 72 h, alamar blue was added to each well (3 µg/mL final concentration) and the plates were incubated for 2 h at 37 °C. The absorbance (540 nM) was evaluated using a Sunrise microplate absorbance reader (Tecan). Cell viability was expressed as the percentage of absorbance of transfected cells compared to untreated cells.

### 2.6. Cell Treatment with Docetaxel and MTT Assay

Hs 578T cells were seeded in 12-well plates 30,000 cells/well and transfected (2X) the day after seeding 200 nM of menin-ASO or control-ASO for 2 times as described above. After 48 h, cells were then treated with 100 nmol/L (half maximal inhibitory concentration (IC50)) of Docetaxel (Sanofi-Aventis, France) for 24 h. MTT (3-(4, 5-dimethylthiazol-2-yl)-2, 5-diphenyl tetrazolium) was added to each well (1 mg/mL final concentration) and the plates were incubated for 2–3 h at 37 °C. Supernatants were then removed and formazan crystals were dissolved in DMSO. The absorbance (595 nM) was evaluated using a Sunrise microplate absorbance reader (Tecan). Each assay was performed in triplicate. Cell viability was expressed as the percentage of absorbance of transfected cells compared to untreated cells.

### 2.7. Cell Cycle Distribution Assay

Hs 578T cells (2 × 10^5^) were seeded in a 100 mm culture dishes. On the following day, the cells were treated with menin or control-ASO 200 nM for 48–72 h. Subsequent to trypsinization and washing with PBS, cells were fixed in 250 µL ice-cold 70% ethanol overnight at 4 °C. The cells were centrifuged (4500 rpm, 10 min, 4 °C), washed with cold PBS and treated with propidium iodide (PI)/RNase staining buffer (BD Pharmingen, San Diego, CA, USA) for 15 min at RT in the dark. DNA content was examined by flow cytometry using a LSRII SORP (Becton Dickinson, France) machine. The percentage of cells in G0, G1, S and G2/M phase was calculated using ModFit software 3.2 (Becton Dickinson, San Diego, CA, USA). The assay was done with triplication.

### 2.8. Cell Apoptosis Assay

Hs 578T cells were plated at the density of 10^5^ cells into 100 mm culture dishes. On the following day, cells were treated with 200 nmol/L of ASO21 or control-ASO twice. After 72 h of incubation, cells were trypsinized, washed with cold phosphate-buffered saline (PBS) and stained using APC Annexin V/Dead Cell Apoptosis Kit with APC annexin V and SYTOX Green for Flow Cytometry (Life Technologies SAS, Courtaboeuf, France). Rates of cell deaths were then measured using FlowJo (Becton Dickinson France SAS, Grenoble, France) The experiments were performed in triplicate.

### 2.9. Assessment of Menin Silencing in TNBC Xenografted Mice

Five-week-old female mice (athymic BALB/C; Charles River Laboratories, Saint-Germain-Nuelles, France) were maintained in the Centre de Recherche en Cancérologie de Marseille (CRCM) animal facility. Approximately 10 × 10^6^ Hs 578T cells were inoculated subcutaneously with 0.1 mL of Dulbecco’s Modified Eagle’s Medium (Life Technologies SAS, Courtaboeuf, France) in the flank region of 5-week-old female athymic BALB/C mice (N = 10). Means of tumor volumes were similar in all groups before therapy. When Hs 578T tumors reached 50 mm^3^ (usually 2 weeks after injection), mice were randomly selected for treatment with menin-ASO or control-ASO. For this experiment, each experimental group consisted of 5 mice and the tumor volume measurements were performed once weekly. After randomization, 12.5 mg/kg menin-ASO or control-ASO were injected intraperitoneally 5 times per week for 12 weeks for ASO monotherapy groups. All mice were routinely observed for signs of systemic toxicity, and body weights were recorded. Tumor size was measured weekly with a caliper in three perpendicular dimensions (x = width, y = length, z = depth). Tumor volume (mm^3^) was calculated as length × width × depth × 0.5236. Data points were expressed as average tumor volume levels ± SEM. All animal procedures were performed in accordance with protocols approved by French law, following the European directives, and with appropriate institutional certification. In particular, Dr. Palma Rocchi has a personal agreement (#A13-477) for the animal handling and experimentation for this study. The mice used in the study were maintained in the animal facility (agreement #13.2700).

### 2.10. Immunoprecipitation and Mass Spectrometry Analysis

The volume of 1 mL of the protein extract from the Hs 578 cells at the concentration of 4 mg/mL was precleaned with 40 μL of the protein A Sepharose (nProtein A Sepharose 4 Fast Flow, REF.17-5280-01, GE Healthcare, Buckinghamshire, UK), and incubated with 5 μg of the rabbit anti-menin polyclonal antibody (Bethyl Laboratories Inc., Montgomery, TX, USA) overnight at 4 °C. The immunoprecipitated complexes were captured by incubating with 40 μL of protein A Sepharose beads for 1 h, 4 °C. Subsequently, the beads were washed 3 times using the lysis buffer, and suspended with 30 μL Laemmli sample buffer 4X, heated at 95 °C for 5 min. Prior to MS analysis, 10% of samples were subjected to WB and silver staining as described [28]. Samples were analyzed by liquid chromatography (LC)-tandem MS (MS/MS) using a Q Exactive Plus Hybrid Quadrupole-Orbitrap online with a nanoLC Ultimate 3000 chromatography system (Thermo Fisher Scientific, San Jose, CA, USA). Relative intensity-based label-free quantification (LFQ) was processed using the MaxLFQ ^2^ algorithm from the freely available MaxQuant computational proteomics platform, version 1.6.2 [29]. Analysis was done on biological triplicates, each injected three times on mass spectrometers. The mass spectrometry proteomics data have been deposited to the ProteomeXchange Consortium via the PRIDE [30] partner repository with the dataset identifier PXD024176.

### 2.11. Bioinformatics Analyses

Functional enrichment analysis was performed using g:Profiler [31] with default setting, threshold *p* = 0.05. The PPI (protein−protein interaction) network of the identified menin’s interactome was built using stringdb webserver V11.0 (Multiple Proteins, Homo sapiens, https://string-db.org/, accessed on 20 June 2020).

### 2.12. Statistical Analysis

Statistical analysis was performed using the GrapPad Prism program (GraphPad, San Diego, CA, USA). All the results were expressed as mean ± SEM. Significance of differences was assessed by a two-tailed Student’s *t*-test. * *p* ≤ 0.05 was considered significant, with ** *p* ≤ 0.01 and *** *p* ≤ 0.001.

## 3. Results

### 3.1. Expression of Menin in Different Breast Cancer Cell Lines

We first assessed menin expressions in 16 TNBC cell lines by Western blotting analysis. As shown in Figure 1, menin is expressed in 6 out of 16 cell lines with the strongest expression in the Hs 578T cell line (Figure 1). Due to the high expression of menin, and limited efficient therapies in TNBC, Hs 578T cells were selected for further evaluation of the effect of menin silencing using ASO technology.

### 3.2. Design and Screening of Menin-ASOs

A set of ASOs against menin mRNA were designed using an institutional tool (see materials and methods). Among them, 32 ASOs (fully phosphorothioate (PS) modified) (from ASO6 to ASO91, Appendix A) were evaluated in the TNBC Hs 578T line at 100 nM concentration. As shown in Figure 2, ASO21 was the most efficient ASO for silencing menin and achieved 60% inhibition of menin protein’s expression. Further WB analysis revealed that ASO21 downregulated menin in a dose-dependent manner (Figure 2b); therefore, ASO21 was used for further analyses in the study.

### 3.3. ASO21 Reduces Hs 578T Viability and Enhances Chemosensitivity of TNBC

We next evaluated the cellular toxicity effect of menin knockdown induced by ASO21. Hs 578T cells were treated with ASO21 or control-ASO (scrambled, SCR) at various concentrations (200 nM, 300 nM, 400 nM). ASO21 decreased the number of viable cells in a dose-dependent manner, with the cell viability rate of control-ASO and ASO21 reduced to 85.30 ± 0.7 and 43.70 ± 0.23 at 300 nM, respectively (Figure 3a,b). Moreover, menin downregulation by ASO21 could sensitize the Hs 578 cells to chemo drugs such as docetaxel, as shown in Figure 3c.

### 3.4. ASO21 Increases Tumor Cell Apoptosis via the Intrinsic Pathway

To decipher the putative mechanism of cytotoxicity induced by ASO21-driven menin silencing, we analyzed cell cycle phases through assessing the DNA amounts by staining DNA with propidium iodide (PI). The percentage of sub-G0-phase death cell population were 9.9% in the untreated group, 13.0% in the control-ASO transfected cells, and 73.3% in ASO21-treated cells (Figure 4a,b). We further evaluated the effect of ASO21 on apoptosis by using an Annexin V-FITC probe to assess the percentages of cells undergoing apoptosis. As shown in Figure 4c, ASO21 significantly induced apoptosis of Hs 578T cells compared with the control. To determine the ASO21-triggered apoptotic pathway, Western blot analysis of caspase-3 (initiator caspases), poly (ADP-ribose) polymerase-1 (PARP-1) and their cleaved forms were performed. After treatment with menin-ASO (200 nM) for 72 h, the levels of cleaved caspase-3 (active form that cleaves PARP) and cleaved-PARP, the hallmark of apoptosis (inactive form, leading to impaired DNA repair) were remarkably increased compared with the control samples in both Hs 578T and MCF-10A cells (Figure 4d), confirming that menin silencing by ASO21 can induce tumor cell apoptosis. These results also suggest that menin inhibition can trigger the intrinsic apoptosis via caspase activation and cleavage of PARP.

### 3.5. Menin-ASO (ASO21) Reduces TNBC-Derived Xenograft Tumor Growth

Twelve days following subcutaneous injection of Hs 578T cells, mice were treated with either ASO21 or control-ASO. At week 7, a drastic reduction in tumor growth was observed in the ASO21 arm (7.275 ± 1931 vs. 28.68 ± 5528 mm^3^ in the control arm, *p* = 0.0012) (Figure 5). Mice were all healthy and exhibited no toxicity throughout the entire experiment (data not shown). At week 12, tumors were removed, and weights were recorded. As expected, the average tumor weights matched the tumor volume measurements, and the average tumor weight with ASO21 (0.36 ± 0.15 g) was lower than the average weight with control-ASO tumors (4.28 ± 0.59 g). These results imply that the depletion of menin by ASO21 reduces the proliferation and tumorigenicity of TNBC Hs 578T derived-xenografts.

### 3.6. Menin’s Interactome Implicates New Putative Functions in TNBC Hs 578T

To further explore the functions of menin in breast cancer, we analyzed the interactome landscape of menin in Hs 578T cells using the IP coupled LC/MS/MS approach. Forty-six potential menin-interacting proteins were found (Figure 6a, Appendix A), two of them, RBBP5 and KMT2B, are well-documented in the literature as menin-interacting proteins (Figure 6a).

Functional enrichment analysis showed that the most representative functions of menin interactors were involved in RNA 3′-end processing (Figure 6b). Menin was found to be engaged in different protein complexes annotated on the CORUM database, indicating the wide diversity of menin’s oncogenic functions in Hs 578T cells (Figure 6c,d, Table 1). The present menin’s interactome identified a menin-associated histone methyltransferase complex, a chromatin regulatory network that drives promoter activation through H3K4me3 (Figure 6c,d). Menin was suggested to be at the crossroad of the polyadenylation complex and BARD1-BRCA1-CSTF complex via interaction with the three subunits of the cleavage stimulation factor (CSTF1, CSTF2, CSTF3), highlighting its potential role in the regulation of mRNA 3′-end process (Figure 6d). Moreover, the menin interactome also suggested the involvement of menin in apoptosis through its interaction with the IRF2BP2-IRF2BP1-IRF2BPL complex (also called DIF-1 complex) which negatively controls the apoptosis.

## 4. Discussion

The *MEN1* gene product, menin, was shown to play a dual role in breast cancer, an anti-proliferative and tumor suppressive function in the normal mammary epithelium and an oncogenic role in sporadic ER-positive breast cancers [16]. Our study provides new insights into the oncogenic functions of menin in TNBC and demonstrates the therapeutic potential of ASOs against menin. Among 16 analyzed TNBC cell lines, menin’s expressions were observed in six lines with the highest level found in the TNBC Hs 578T cell line (Figure 1). To assess the effect of menin silencing using ASO technology, we chose Hs 578T cells due to their menin overexpression profile and because they are derived from a TNBC, a tumor with unfavorable prognosis and limited therapeutic options [2].

The ASO technology has been adapted for targeting the oncogenic functions of intracellular menin, which act via either a co-activator or a co-repressor effect of transcriptional activity. The use of small molecules for disrupting some protein−protein interactions is an alternative option, such as inhibitors of menin-MLL1 interaction, but this approach remains limited due to several challenges such as the bivalent interaction of menin and MLL, and the wide diversity of menin interactors [11]. ASOs are relatively easy to synthetize and transferable into clinical applications [25]. During the past seven years, several ASOs have been approved for clinical usage for homozygous familial hypercholesterolemia (Kynamro), spinal muscular atrophy (Spinraza), Duchenne muscular dystrophy (EXONDYS 51), polyneuropathy of hereditary transthyretin-mediated (hATTR) amyloidosis (Tegsedi and Batten disease (Milasen) [32].

We found that one ASO specifically silenced menin gene expression and reduced menin levels by 60% at 100 nM with an increased inhibition effect at higher concentrations. Menin silencing decreased cell growth in vitro and enhanced apoptosis *via* the intrinsic apoptotic pathway since the caspase 3-PARP apoptotic pathway was activated (Figure 4c,d). Consistent with these in vitro data, the systemic administration of menin- ASO monotherapy suppressed the Hs 578T-derived xenograft tumor growth in mice. The results from menin’s interactome in Hs 578T cells provides new clues for understanding the relationships between menin and apoptosis.

Apoptosis is a critical function of menin, although the detailed mechanism of actions remains largely unknown. In murine embryonic fibroblasts (MEFs), the overexpression of menin promotes apoptosis via a BAX/BAK-dependent apoptotic pathway, and loss of menin results in the upregulation of procaspase 8 at both mRNA and protein levels [33]. It was also reported that menin could epigenetically regulate the transcription activity of *caspase 8* through binding to the 5’-untranslated region (5’-UTR) of the *Caspase 8* locus [34]. In addition, upon genotoxic stress, such as irradiation, the overexpression of menin amplifies activations of capase-3 and p21, and increases p53 acetylation, thereby promoting apoptosis in endocrine pancreatic tumor cells INS-r3 [35]. These findings imply that menin might regulate apoptosis in a context-dependent manner, through both the intrinsic and extrinsic apoptotic pathways.

Here we show that ASO-driven menin knockdown induces cell programmed cell death via caspase-3 activation and induction of PARP cleavage. Interestingly, we found that menin could modulate apoptosis via interacting with IRF-2BP2 (interferon regulatory factor-2 binding protein 2) that was identified as an apoptotic protein and its knockdown leads to apoptosis of breast and prostate cancer cells but not of other cell types [36]. IRF-2BP2 is a transcriptional repressor of the proapoptotic gene *FASTKD2* (Fas-activated serine-threonine kinase domain 2) in breast cancer cells [37]. *FASTKD2* encodes a protein that is localized in the mitochondrial inner compartment and plays a role in mitochondrial apoptosis. Nonsense mutations of *FASTKD2* have been reported to result in cytochrome c oxidase deficiency [38]. Thus, it is expected that inhibition of IRF-2BP2 could lead to increased expression of *FASTKD2* with subsequent apoptosis induction. Menin could also play a role in apoptosis through recruitment of various chromatin regulatory enzymes such as SIRT1, a histone deacetylase enzyme, that facilitates IRF-2BP2 antiapoptotic function. Both IRF-2BP2 and menin were previously shown to be associated with SIRT1 in gene expression modulation [39]. Menin was previously shown to regulate apoptosis through transcriptional modulation of pro-apoptotic genes, such as caspase 8 [34]. Interestingly, menin was found to be associated with TSS region of FASTKD2 in LNCaP cells in our Chip-seq analysis (personal data, under submission). All together, these observations suggest that menin might bind to IRF-2BP2 and suppress expression of proapoptotic genes in TNBC such as a pro-apoptotic factor FASTKD2, resulting in promotion of cellular survival, and depression of apoptosis.

Results of the interactome analysis suggest a novel role of menin in polyadenylation via interaction with the cleavage stimulation factor (CSTF) which is a part of the polyadenylation and BARD1-BRCA1-CSTF complexes, as found in our functional enrichment analysis with CORUM database (Figure 6c,d). The polyadenylation process that occurs during transcription (e.g., addition of the polyA at the pre-mRNA tail) plays an important role in mRNA stability, localization, and translation. Menin-interacting proteins CSTF facilitate the recognition of the polyadenylation signals (PAS) by another protein complex named cleavage-polyadenylation specificity factor (CPSF) [40]. Alternative polyadenylation (APA) refers to the production of multiple transcripts with variable 3′UTR length from one locus gene due to the presence of multiple PAS in pre-mRNA [41]. Numerous studies have provided evidence showing that APA is strongly associated with cancer and can serve as a novel prognostic biomarker [42,43]. Two recent analyses have shown the association of various 3′UTR shortening transcripts in TNBC, tumor progression and decreased prognosis [41,44]. The shortening of 3′UTRs enables key genes to escape microRNA repression, thus leading to higher expression and promoting proliferation [41], [45,46,47]. Menin was also found to interact with both initiating (pSer5) and elongating (pSer2) forms of RNAPII during transcription [21,48]. RNAPII was is associated with APA machinery and regulates APA function [49]. Analysis of GO-BP enrichment also suggests the involvement of menin in transcription termination through interaction with CSTF complex (Appendix A). All together, these new data suggest that menin could play an important role in polyadenylation, leading to a regulation of a set of key genes involved in TNBC progression.

We also found that in vitro menin silencing could act synergistically with docetaxel in TNBC (Figure 3c). The combination of menin silencing with the DNA-crosslinking platinum chemotherapies could be of particular interest in TNBC as deficient DNA damage repair is a biological hallmark of some TNBCs [50]. At present, the administration of platinum as single agent chemotherapy in metastatic TNBC is associated with a median progression-free survival of 3.1 months and an overall response rate of 31%. Over the past 6 years, new therapeutic agents have been approved by the US Food and Drug Administration (FDA), including inhibitors of PARP for TNBC without germline BRCA1/2 mutations. Therefore, an attractive option in TNBC would be to promote apoptosis with menin-ASO in combination with chemo (induce DNA damages) or PARP inhibitors (inhibit DNA repair process). The evaluation of the optimal combinations and treatment sequencing between systemic treatments and RNA therapeutics remains to be established by further preclinical experiments.

ASO-guided menin knockdown has been demonstrated in the Hs 578T cell line as a proof-of-concept study. It is, however, anticipated that this effect could be observed only in menin-expressing TNBC as it has been described for castration-resistant prostate cancer cell lines that do not express an androgen receptor such as PC-3 (C.Cherif, Oncogene, in revision). Further basic research experiments including other TNBC models would enable this therapeutic strategy to be more strongly supported.

## 5. Conclusions

Collectively, this study supports the hypothesis that menin silencing can trigger apoptosis and inhibit tumor growth in TNBC and provides a rationale for the use of ASO-based therapeutics against menin as a monotherapy or in combination with chemo or PARP inhibitors in menin-positive TNBC.

## Figures and Tables

**Figure 1 biomedicines-09-00795-f001:**
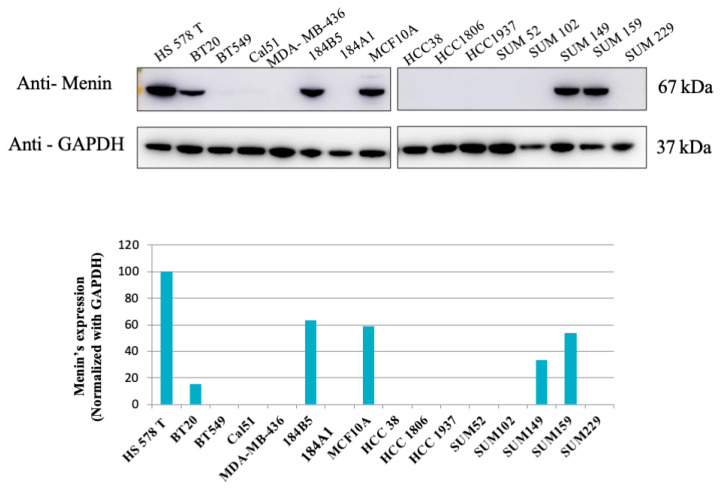
Western blot analysis of Menin expression in 16 TNBC cell lines. The expression of menin in 16 TNBC (triple-negative breast cancer) cell lines were evaluated using WB analyses. Six out of sixteen cell lines displayed menin’s expression, and Hs 578T had the highest level of menin. Bands were quantified by densitometry and menin was normalized to GAPDH levels.

**Figure 2 biomedicines-09-00795-f002:**
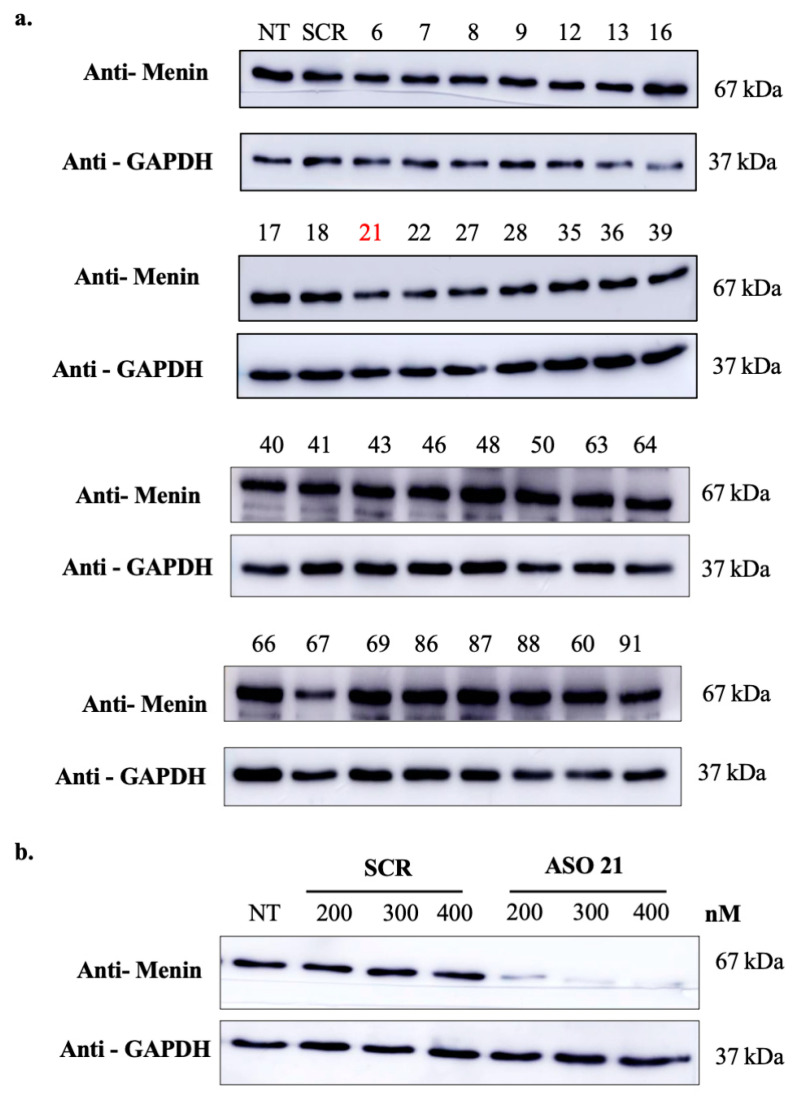
Screening of menin-ASO inhibiting menin’s expression in TNBC. (**a**) Screening of menin-targeting ASOs. Hs 578T cells were treated with 100 nM of different menin-ASOs or control-ASO (SCR). Three days after the second transfection, proteins were extracted and analyzed by Western blot. ASO21 exhibits high potency of menin inhibition. (**b**) ASO21 inhibits menin’s expression in dose-dependent manner. Hs 578T cells were treated with ASO21 or control-ASO (SCR) at different concentrations: 200 nM, 300 nM and 400 nM. Three days after the second transfection, proteins were extracted and analyzed by Western blot.

**Figure 3 biomedicines-09-00795-f003:**
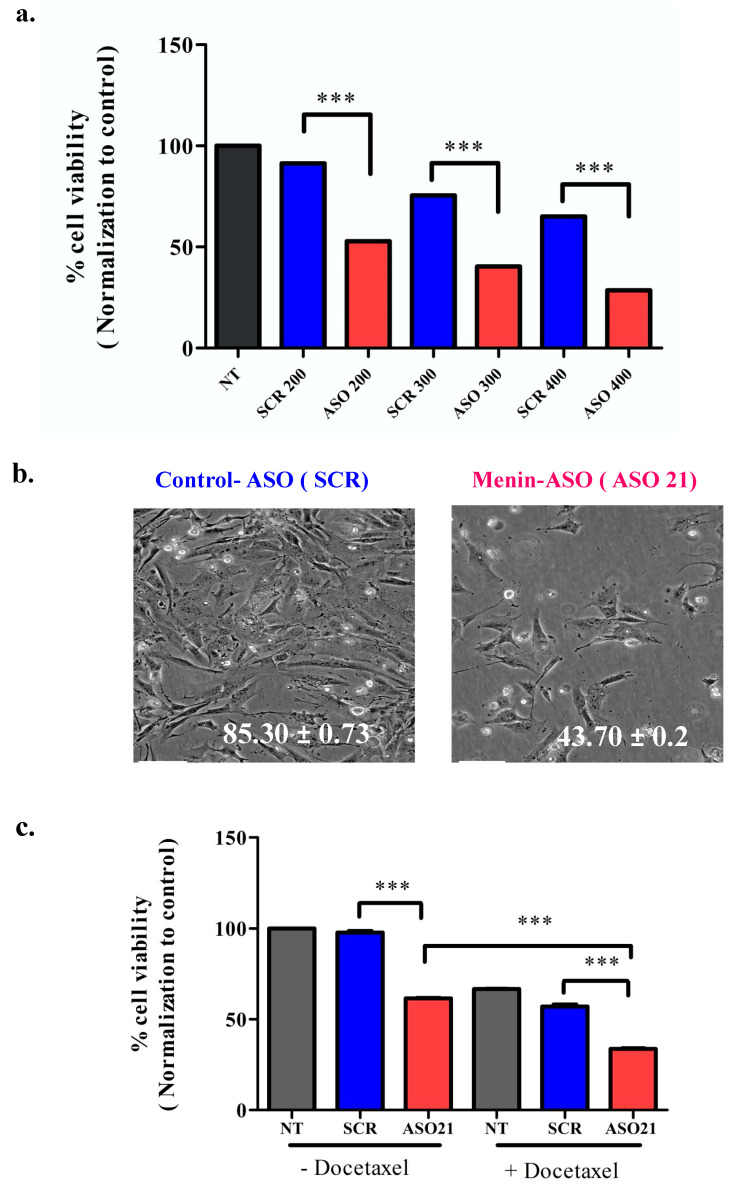
ASO21 reduces Hs 578T viability and enhances chemosensitivity of TNBC. (**a**) ASO21 decreases Hs 578 cellular viability in dose dependent manner. Hs 578T cells were treated with ASO21 and scrambled-ASO (SCR) at various concentrations (200 nM, 300 nM, 400 nM). Three days after the second transfection, cell proliferation was assayed by alamar blue. (**b**). Representative images of Hs 578T cells that were treated with control-ASO or menin-ASO (ASO21). (**c**) ASO21 enhances chemosensitivity of Hs 578T cell to Docetaxel. After the second ASO transfection at 200 nM 2 days, Docetaxel 100 nM was added and incubated for 2 days before the MTT tests. Menin knockdown using ASO 21 increased cellular sensitivity to Docetaxel since cell survival decreased significantly with combination of this chemodrug with menin-ASO compared with monotherapy using menin-ASO alone was considered significant, with *** *p* ≤ 0.001.

**Figure 4 biomedicines-09-00795-f004:**
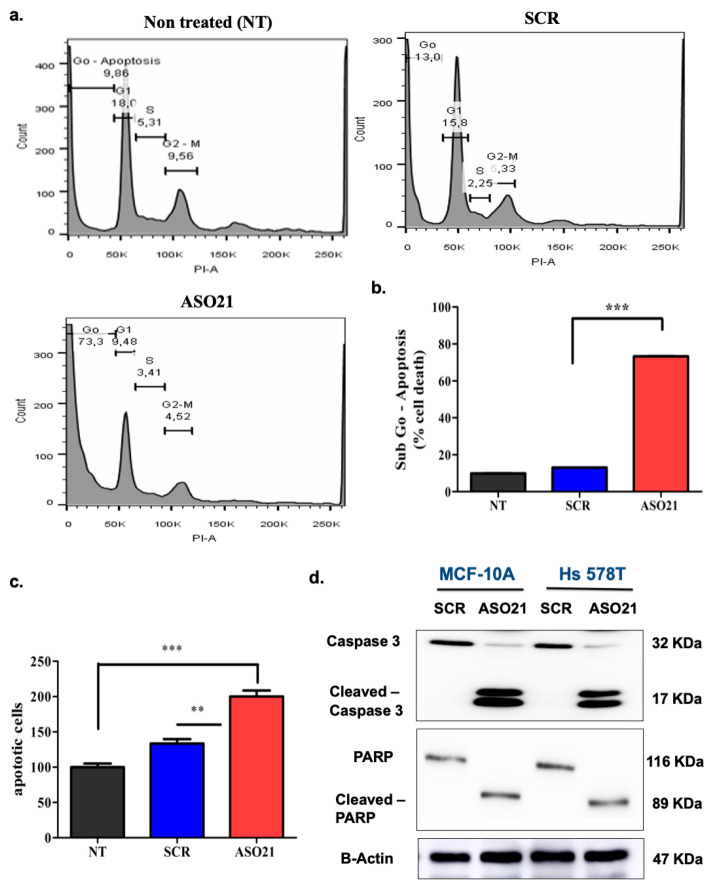
ASO21 induces apoptosis via the intrinsic pathway in Hs 578T cells. Hs 578T cells were treated with ASO21 and SCR at 200 nM. Three days after second transfection, cells were harvested for Western blotting analysis or for cell apoptosis and cell cycle analyses using Annexin V and propidium iodide (PI). (**a**) The plots of cell cycle profiles of nontreated (NT), SCR-transfected cells (SCR) and ASO21-transfected cells (ASO21). (**b**). The graph showing the percentages of cells in sub-Go phase or death cells in NT, SCR and ASO21 samples. ASO21 caused a remarkably high proportion of death cells (73.3%) compared with NT (9.86%) and SCR (13%). (**c**) The graph displays the percentages of apoptotic cells in NT, SCR and ASO21 samples. ASO21 increased apoptosis in Hs 578T cells with the increased number of cells undergoing apoptosis compared with the control. (**d**) ASO 21 increased cleavage of caspase-3 and PARP in TNBC. Appearances of cleaved caspase-3 and cleaved–PARP in Hs 578T or MCF-10A treated with menin-ASO (ASO21) confirm that menin inhibition by ASO21 induces apoptosis and suggest that ASO21 can trigger the intrinsic apoptotic pathway.** *p* ≤ 0.01 and *** *p* ≤ 0.001.

**Figure 5 biomedicines-09-00795-f005:**
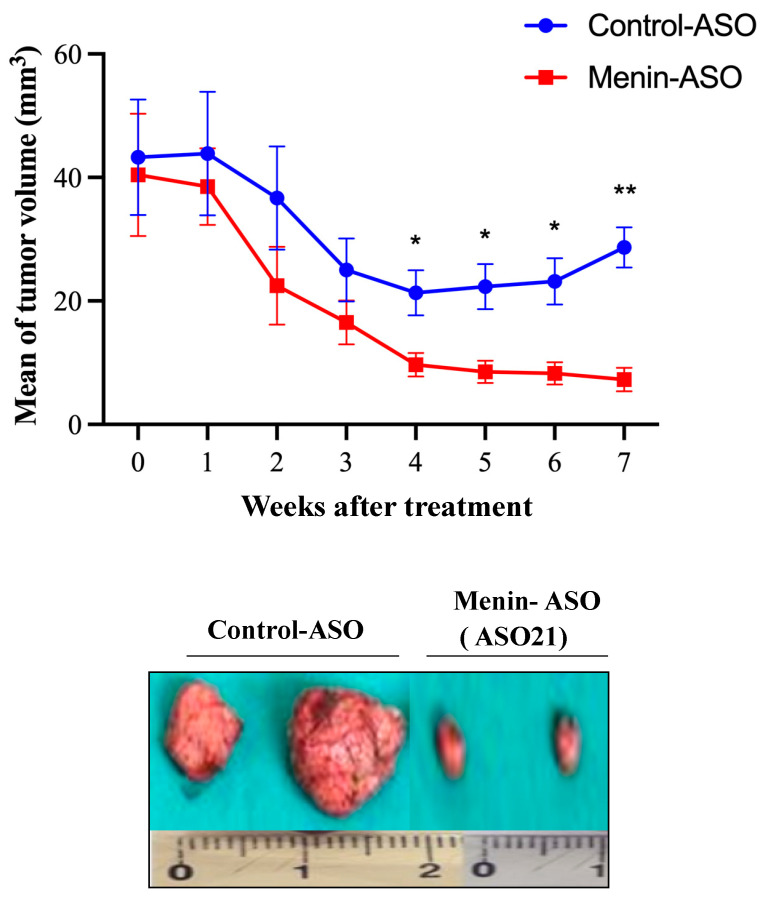
Menin-ASO (ASO21) inhibits Hs 578T- derived xenograft growth. Approximately 10 × 10^6^ Hs 578T cells were inoculated subcutaneously in the flank region of 5-week-old female athymic BALB/C mice. Means of tumor volumes were similar in all groups before therapy. When Hs 578T tumors reached 50 mm^3^, 5 mice were randomly selected for treatment with 12.5 mg/kg menin-ASO or control-ASO. ASOs were injected intraperitoneally 5 times per week for 12 weeks for ASO monotherapy groups. Tumor size was measured weekly with a caliper in three perpendicular dimensions (x = width, y = length, z = depth). Tumor volume (mm^3^) was calculated as length × width × depth × 0.5236. Data points were expressed as average tumor volume levels ± SEM. Comparison of menin-ASO and control-ASO indicated that knockdown of menin by ASO inhibits TNBC tumor progression in vivo. * *p* ≤ 0.05 was considered significant, with ** *p* ≤ 0.01.

**Figure 6 biomedicines-09-00795-f006:**
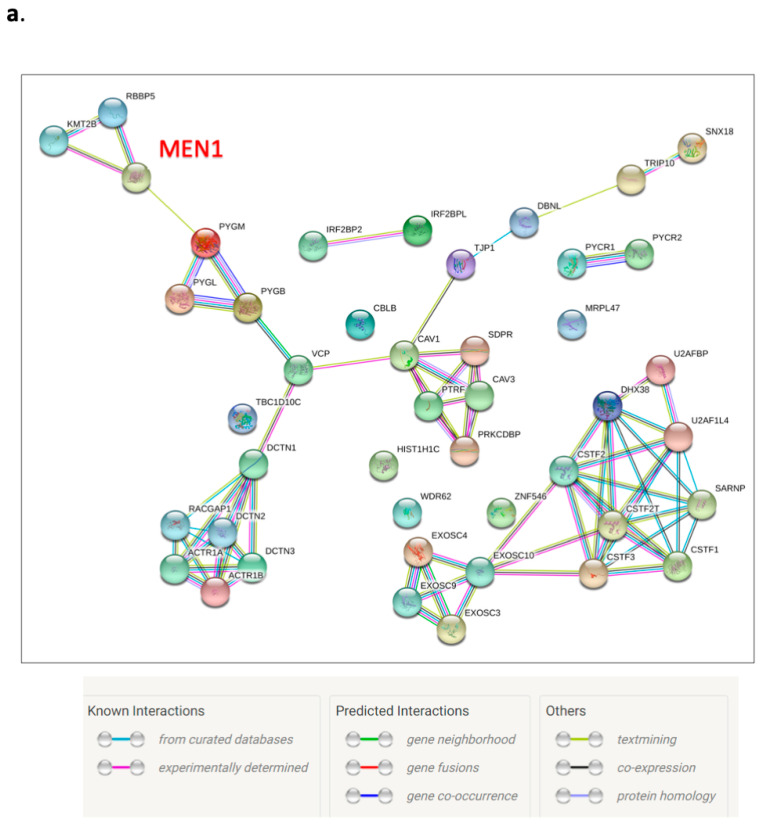
Identification of Menin’s interactome in Hs 578T cells using IP coupled LC/MS/MS. (**a**) The PPI network of identified menin’s interactome was constructed by stringdb websever. (**b**,**c**) Functional enrichment analysis of menin’s interactome using g:Profiler (https://biit.cs.ut.ee/gprofiler/gost, accessed on 30 December 2020); (**b**) top ten GO-BP terms enriched racking by *p*-value; (**c**) enriched protein complexes found in the CORUM database. (**d**) Modeling of protein complexes found in the menin’s interactome. Green nodes: proteins found in identified menin’s interactome in TNBC. Grey nodes: the ones not found in the interactome.

**Table 1 biomedicines-09-00795-t001:** Functions of menin-associated protein complexes enriched in CORUM database (https://mips.helmholtz-muenchen.de/corum/).

Term_Name	Term_Id	Functions
**Cleavage stimulation factor**	CORUM:1146	Cleavage step, mRNA 3′-end processing (polyadenylation)
**BARD1-BRCA1-CSTF complex**	CORUM:2211	Interaction of BARD1 with CstF inhibits RNA polyadenylation and ensures that at DNA damage sites nascent RNAs are not erroneously polyadenylated.
**Polyadenylation complex** (CSTF1, CSTF2, CSTF3, SYMPK CPSF1, CPSF2, CPSF3)	CORUM:1147	mRNA 3′-end processing (polyadenylation)
**Menin-associated histone methyltransferase complex**	CORUM:1254	chromosome remodeling, epigenetic regulation
**IRF2BP2-IRF2BP1-IRF2BPL complex**	CORUM:6848	regulation of apoptotic process, negative regulation of transcription
**Exosome**	CORUM:7443	degradation and maturation of a wide variety of RNAs

## Data Availability

The mass spectrometry proteomics data presented in this study are openly available in the ProteomeXchange Consortium with the dataset identifier PXD024176. The data presented in this study are available on request from the corresponding author.

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
