# Peer review of "Antisense Oligonucleotide-Based Therapeutic against Menin for Triple-Negative Breast Cancer Treatment"

_biomedicines, 2021, doi:10.3390/biomedicines9070795_

Round 1

Reviewer 1 Report

The manuscript describes the effects of menin downregulation by antisense oligonucleotide in TNBC cell line Hs578T and post-treatment reduction of  xenograft tumors of the same cell line in mice. The effects of menin downregulation are strong and statistically significant; however, the manuscript would benefit from providing more explanations for the issues that are listed below.

Here are major concerns this research should address:

  1. The important information about ASO used is missing. As unprotected all-phosphodiesther oligonucleotides are nuclease-sensitive, and their degradation products might be toxic, the authors need to clarify if the ASO backbone was straightforward DNA, or it was chemically modified for nuclease resistance and to facilitate cellular uptake? If it was not modified, the issues mentioned above need to be discussed.
  2. In the mice experiment, gene knockdown should be confirmed in target tissues.
  3. In the Results, as well as in the Discussion, authors are generalizing the effect of one ASO studied in one TNBC cell line, to make the conclusions about all TNBCs. However, more than half of the cell lines they screened initially, did not even express menin; and for all consequent experiments the cell line with the highest menin expression was chosen. Can the authors expect the same effect of the ASO used, in all TNBCs? Either additional experiments should be performed, or Results and Discussion should be re-written accordingly.

Reviewer 2 Report

The authors examined significance of menin inhibition with oligonucleotide technology. This is a well-written and well-structured study, which has multiple strong sides and several weaknesses.

Strong sides – use of oligonucleotides with the purpose of pursuing an animal study, test of multiple oligonucleotide to identify best acting candidate, demonstration of successful use of oligonucleotides in vivo.  Weakness – use of one cell line to test function of ASO21, use of double transfection to deliver oligonucleotides and lack of confirmation with an additional method, for example, siRNA transfection, importance of menin knockdown in TNBC. Oligonucleotides have benefit over siRNA when used in vivo, siRNA is a better technology when used in vitro. Use of single cell line and one method to confirm menin function keeps the possibility of non-specific pro-apoptotic function of this particular sequence in this particular cell line. These questions can be addressed in the future study when authors examine aso21 oligonucleotide in additional TNBC cell lines.

Overall, it’s a well-written study, which will be of interest to readers.

Please make following changes or corrections to the manuscript:

  1. Clarify the use of double transfection in the “Materials and Methods” section. Two experimental descriptions are currently mixed in the “Transfection with ASOs” section. Also, please explain in the same sections why cells were transfected twice
  2. Please use consistently the cell line name Hs 578T in the document (several instances of Hs578T)
  3. Clarify in either figure legend or manuscript text what concentration of aso21 was used in Figure 3c.
  4. Correct use of two periods in the figure 4 legend: “ASO21 induces apoptosis via intrinsic pathway in Hs 578T cells. .Hs 578T cells”.
  5. Correct in Figure legend 5 what appears as extra spaces in the pdf document: “caliper in  three”
  6. Correct in Figure 5 legend use of cube mm as superscript and -. “50 mm3, 8 mice were randomly selected for treatment with Menin ASO or control-.12.5”

Reviewer 3 Report

The work presented by Nguyen et al. is well descripted and written.

However, some improvements are needed:

1) the data have been produced in just 1 TNBC cell line. In order to be more impressive, i suggest to confirm some data also in 1 more TNBC cell line among those that expresses Menin. The exps to produce are those in Fig. 2b, fig 3, fig 4 c-d.

2) Since the chemotherapy is still the standard of care in TNBC, in my opinion the most relevant aspect of the paper is the synergistic effect on apoptosis of ASO21 in combination with docetaxel. So i suggest to add apoptosis assays in the group Docetaxel+ASO21 (fig 4)

3) then, just the remark that it would have been more interesting to have results about Docetaxel+ASO21 combination in vivo 

Round 2

Reviewer 1 Report

Minor edits have been introduced into the manuscript after the first round of reviews, although all the reviewers described major flaws that should be addressed.

Reviewer 3 Report

I indestood the point of view of the authors and their explaination to my request.

Round 3

Reviewer 1 Report

Thank you for introducing changes in the text of the manuscript. Yes, the major edits should include additional experiments; without them data is incomplete to fully support the hypothesis. In my opinion, in the present form the manuscript is below acceptable scientific level of the journal.
